# Cross-cultural adaptation and psychometric evaluation of the urdu version of the morisky, greene, and levine medication adherence scale (MGLS-4) for major depressive disorder patients

Sohail Riaz[1], Fazli Khuda[1]*, Nadia Shamshad Malik[2], Nitasha Gohar[2], Ayesha Rashid[3], Abuzar Khan[1], Abdur Rahman[4], Sajid Ali[5], Asif Jan[1], Aqeel Nasim[6]

1 Department of Pharmacy, University of Peshawar, Peshawar, Pakistan, 2 Faculty of Pharmacy, Capital University of Science and Technology, Islamabad, Pakistan, 3 Department of Pharmacy, The Women University Multan, Multan, Pakistan, 4 Department of Psychology, International Islamic University, Islamabad, Pakistan, 5 Department of Biotechnology, Abdul Wali Khan University, Mardan, Pakistan, 6 Balochistan Institute of Nephrology Urology Quetta BINUQ, Quetta, Pakistan

* fazlikhuda@uop.edu.pk

## Abstract

In Pakistan, Major depressive disorder (MDD) contributes significantly to the mental health burden. It is crucial to understand patients' medication adherence status for developing a strategy for improving adherence and treatment outcomes. Therefore, a valid and reliable tool in the local Urdu language is required. The Morisky, Greene, and Levine Medication Adherence Scale (MGLS-4) is a reliable, valid and straight-forward instrument to assess medication-taking behavior. The valid and reliable Urdu translation of MGLS-4 can fill this gap within the local context. Therefore, the present study aims to validate the Urdu Morisky, Green and Levine Adherence Scale (UMGLS-4) for MDD patients. This was a quantitative, cross-sectional validation study for Pakistani MDD patients. Reliability was measured using Cronbach's α and for test-retest reliability intraclass correlation coefficient (ICC) was calculated. Validity was assessed through face validity, content validity, construct validity, and convergent validity with the Drug Attitude Inventory (DAI-10). Descriptive and inferential statistical analyses were carried out to demonstrate adherence level and statistical significance, respectively. Linear regression was applied to find the association between the UGMLS-4 score and demographic characteristics. The UMGLS-4 demonstrated high reliability (Cronbach's $\alpha = 0.829$) and a significant strong ICC (x = 0.601, $p < 0.01$) was detected. Exploratory factor analysis (EFA) revealed a single-factor structure explaining 66.084% of the variance. Confirmatory factor analysis (CFA) confirmed good model fit (GFI = 0.950, AGFI = 0.920, NFI = 0.930, RMSEA = 0.050, SRMSR = 0.055). Medication adherence was observed to be high in 39.1% of patients, moderate in 28.6%, and poor in 32.3%. Significant associations were found between adherence scores and gender, educational attainment, and occupational status ($p < 0.005$) with

**Data availability statement:** All relevant data are within the article and its Supporting Information files.

**Funding:** The author(s) received no specific funding for this work.

**Competing interests:** The authors have declared that no competing interests exist.

education predicting adherence (B = 0.301, $p < 0.000$), indicating the scale's robustness in detecting adherence variations among Urdu-speaking MDD patients. The UMGLS-4 is a reliable and valid tool for assessing medication adherence in Pakistani MDD patients, effectively capturing adherence variations across demographic variables.

## Introduction

Major Depressive Disorder (MDD) is a significant public health issue and recognized as the leading cause of disability worldwide by the WHO, affecting over 300 million people globally [1]. It's characterized by persistent sadness and a lack of interest in life, affecting individuals' daily functioning and contributing to a high societal and economic burden [2]. The prevalence of MDD and its association with chronic diseases highlight the urgency of addressing it through effective public health strategies [3].

In Pakistan, over 20 million individuals (10% of the population) have a mental illness [4], with 4.8 million comprising MDD [5]. The prevalence of depression has risen from 3.1% in 2009 to 3.8% in 2019. In 10 years, the prevalence has increased from 2.95% to 3.14% [6]. MDD, the most prevalent mental ailment, also imposes high economic and productivity costs on society. Pakistan saw the most significant increase in DALYs for MDD (89.4%) between 1990 and 2016 [7]. However, these findings do not depict the current scenario of MDD prevalence in Pakistan. For example, cross-sectional research conducted by telephone survey with 820 participants from three major cities found that the average Prevalence of MDD in urban regions was 46% [8]. Another study conducted in 2015 reported that 32% of patients had MDD [9]. These findings demonstrate iceberg pheromone regarding MDD prevalence in Pakistan. Only the cases that are reported are considered. At the same time, MDD cases that are diagnosed but not reported, cases that lack follow-up, and cases that are not diagnosed due to a lack of social & family support are not counted.

Adherence can be explained as "the extent to which an individual follows the recommendations highlighted by a healthcare professional" Any deviation from this is called Medication Nonadherence (MNA) [10]. Studies demonstrated that MNA ranged from 15 to 90% among individuals with MDD [11]. At the same time, some studies suggest that patients suffering from depression are three times more likely than the other diseased population to have poor medication adherence and poor health behavior [12]. This difference in adherence rates may represent challenges to medication adherence or the effect of confounding variables such as age, socioeconomic position, and illness duration. In addition to the factors responsible for MNA among chronic disease patients, many other factors greatly influence the medication adherence of MDD patients, such as suicidal thoughts, pessimism, impaired cognition, lower self-esteem, side effects of ADs, and excessive melancholic moods [13]. MNA typically increases medication doses, resulting in more significant health expenditure, serious adverse events, misdiagnoses, unnecessary therapy, disease aggravation, and mortality [14]. These factors substantially affect patients and the

health sector, such as increased healthcare expenses [15]. Thus, assessing the patient's medication adherence and opting for relevant interventions to improve it could be crucial.

Given the profound impact of MNA on MDD patient outcomes, there is a need for screening tools to monitor and enhance adherence to treatment regimens [16]. The Morisky, Greene, and Levine Medication Adherence Scale (MGLS-4) (Also known as Medication Adherence Questionnaire) emerges as an instrument, offering a reliable and straightforward method to assess medication-taking behavior [17,18].

The MGLS-4 was developed in 1986 from a 5-item version and has been validated for reliability (Cronbach $\alpha = 0.61$) in a study with 290 hypertensive patients. This 4-item scale evaluates if patients forget, are careless with medication, or stop taking it when feeling better or worse. Responses are scored from 0 ("No") to 1 ("Yes"), with total scores categorizing adherence as high (0 "yes"), moderate (1–2 "yes"), or low (3–4 "yes") [19]. It has been extensively validated across diverse languages and cultural contexts, including Arabic [20,21], Chinese [22], Indonesian [23], Taiwanese [24] and Singaporean [25] populations. This scale's usefulness is further evidenced by its use in monitoring medication adherence across a spectrum of conditions such as cardiovascular diseases [26–28], acquired immunodeficiency syndrome (AIDS) [29], Diabetes Mellitus (DM) [30], Pregnancy [31], cancer [32–34], Chronic Obstructive Pulmonary Disease (COPD) [35] and mental disorders [36–38].

In the context of Pakistan, MGLS-4 is widely used for various health conditions, including cardiovascular diseases [39–41], DM [42], Kidney failure [43] and mental health disorders such as bipolar disorder [44], and schizophrenia [45]. However, its utility within the Pakistani population hinges on its cultural and linguistic adaptation, ensuring its relevance and applicability to Urdu-speaking patients. Despite its widespread use, there is a notable lack of evidence validating MGLS-4 within the Pakistani context. This gap highlights the need for studies that examine the scale's effectiveness and cultural appropriateness for Pakistani patients. By validating the MGLS-4 for use in Pakistan, healthcare professionals will access a concise and culturally relevant tool to quickly identify at-risk patients, enabling targeted interventions without requiring significant time or resources. The validated tool could be routinely utilized in clinical setups to monitor patient progress, address barriers like forgetfulness or side effects, and enhance treatment outcomes for MDD patients. Hence, the current research seeks to translate, culturally adapt, and validate the MGLS-4 in Urdu for patients with MDD in Pakistan.

## Methodology

### Study design and settings

The study was designed as a quantitative, questionnaire-based, cross-sectional validation study. The setting for this study was the Shaikh Zayed Medical Complex, Lahore, Pakistan. The duration of the study was from March - June 2024.

### Study population and sampling

The study specifically included outpatients diagnosed with MDD by psychiatrists using the DSM-5 criteria. The inclusion criteria required participants to be between 18 and 60 years old, fluent in Urdu and provide informed consent with no cognitive or communicative impairments. Exclusion criteria encompassed patients outside this age range, those lacking Urdu proficiency, individuals diagnosed for the first time, or those who did not sign consent forms. The convenient sampling method was used. Recruitment was carried out in-person by Principal Investigator (S.R) during pre-scheduled outpatient visits at the hospital. A sample size of 120 participants is established by applying the 1:30 item-to-respondent rule for the MGLS-4 survey [46]. An additional 35% is factored in to accommodate potential dropouts, adjusting the total number required to 162 participants.

### Translation and cultural adaptation of morisky, greene, and levine medication adherence scale (MGLS-4) into urdu

The translation and cultural adaptation of the MGLS-4 into Urdu, hereafter referred to as UMGLS-4, followed the procedural guidelines [47].

**Initial or forward translation.** Initially, the MGLS-4 was subjected to translation by two different translators. The first translator (T1) understood the clinical aspects of the MGLS-4, while the second translator (T2) lacked medical literacy and familiarity with the MGLS-4 framework.

**Synthesis of translations.** Following the independent translations, T1 and T2 collaborated with the principal investigator to synthesize their translations into a preliminary UMGLS-4, reaching a consensus through deliberation.

**Back translation.** Subsequently, UMGLS-4 was translated back into English by another pair of translators (BT1 and BT2), neither of whom was aware of the original MGLS-4 or possessed a medical background, ensuring an unbiased translation process.

**Expert committee review.** The content validity of the translations was rigorously assessed by an expert panel convened by the principal investigator. This panel consisted of five members, each bringing distinct expertise relevant to the study's objective: the principal investigator, experienced in psychometric evaluations; a psychiatrist familiar with depressive disorders; two assistant professors of pharmacy, acting as supervisor and co-supervisor, with backgrounds in pharmacotherapy; and a psychologist specializing in cognitive assessment. The equivalence of the UMGLS-4 was evaluated across four dimensions (semantic, idiomatic, experiential, and conceptual). Additionally, committee members were tasked with categorizing each item according to its importance, labeling them as "essential", "useful but not essential", or "not useful". Upon reaching a consensus regarding the translations' accuracy and cultural appropriateness, the definitive Urdu version of the MGLS-4 was finalized for face validity among 20 MDD patients.

## Data collection and questionnaires

Data were collected using paper forms of data collection instrument, which were supervised and collected by principal investigator (S.R) to ensure accuracy and completeness. The data collection instrument was structured into four segments: 1) demographic information, 2) histories of clinical conditions and adherence phase, 3) the UMGLS-4, and 4) the Urdu version of the Drug Attitude Inventory (DAI 10) with Cronbach's $\alpha$ of 0.70 was employed to evaluate convergent validity [48]. The DAI-10, a shortened form of the original 30-item scale, assesses patients' attitudes toward medications through 10 items, scoring from -10 to +10 based on positive or negative attitudes [49]. The data collection was started from 22nd March – 10th June 2024.

## Ethical consideration

The study was conducted in compliance with the Declaration of Helsinki [50]. We received ethical approval from the University of Peshawar, Pakistan's ASRB (ASRB PhD/5th 2023). Permission letters were also granted (Ref:232/PHAR) from the Department of Pharmacy and the Technical and Ethical Review Committee (TERC) of Shaikh Zayed Medical Complex, Lahore (Ref: 02-TER/NHRC-SZH/Ext-Sc/465). Permission from developers of MGLS-4 and Urdu DAI-10 was obtained through email. Participants signed the written informed consent in Urdu. Data obtained was coded and analyzed anonymously.

## Statistical analyses

The descriptive statistics, such as frequencies, percentages, means, and standard deviations, were used to summarize demographic details, clinical characteristics, and adherence levels of MDD patients. The reliability and internal consistency of the UMGLS-4 were assessed through Cronbach's $\alpha$, with an acceptable range of 0.45–0.9 [51]. Test-retest reliability was assessed to ensure consistency in participants' responses using UGMLS-4 administered to 96 participants twice, two weeks apart. Test-retest reliability assessment involved calculating the intraclass correlation coefficient (ICC) and Spearman's rank correlation coefficient (Spearman's r). Spearman r values: ≥0.75 indicate very good to excellent correlation, 0.51–0.75 moderate to good, 0.26–0.50 small, and 0–0.25 little to no correlation [52].

Convergent validity was determined using Pearson's correlation, comparing the UMGLS-4 with the validated Urdu version of DAI-10. Exploratory factor analysis (EFA) utilizing Varimax rotation was conducted to establish construct validity, retaining factors exhibiting eigen values >1 and adopting a factor loading threshold of ≥ 0.4. Before executing confirmatory factor analysis (CFA), Bartlett's sphericity test and the Kaiser-Meyer-Olkin (KMO) measure were applied to assess the data's suitability for factor analysis. Confirmatory factor analyses (CFA) assessed model fit using the Goodness of Fit Index (GFI) and Adjusted GFI, with acceptable thresholds set at ≥0.95 and ≥0.9, respectively. Additionally, the Normed Fit Index (NFI) required a minimum of 0.95 for adequacy. The Root Mean Square Error of Approximation (RMSEA) and the Standardized Root Mean Square Residual (SRMSR) were considered satisfactory with values <0.06 and <0.08, respectively [53].

The Shapiro-Wilk test confirmed the non-normality of the data ($p < 0.05$), justifying the use of non-parametric tests. The Chi-square test/ Fisher's Exact Test was employed for categorical variables. The Kruskal-Wallis and Mann-Whitney U tests were selected for continuous variables. Significance levels were established at $p < 0.05$ for the Chi-square/Fisher's Exact tests and $p < 0.005$ for the Kruskal-Wallis and Mann-Whitney U tests. This methodological approach facilitated a thorough exploration of medication adherence. Both continuous and categorical data analyses were conducted. This dual strategy was chosen to enhance the depth of understanding regarding adherence behaviors in MDD patients.

Regression was used to assess associations between UMGLS-4 scores, MDD patient demographics, and clinical characteristics. Variables that demonstrated significant correlations ($p < 0.05)$ in simple linear regression analyses were incorporated into multivariate linear regression analyses. Multicollinearity was assessed using Variance Inflation Factor (VIF) values, All the statistical analyses were carried out using Statistical Package for Social Science (SPSS) Version 20.0 and Analyses of Moment Structure (AMOS).

## Results

### Demographics & clinical characteristics

A total of 220 MDD patients participated in this study. Among these participants, 50.5% were male (n=111), and 49.5% were female (n=109). The 30–39 year age group was the most represented at 33.6% (n=74). Most participants were married, accounting for 56.4% (n=124), and the most prominent educational subgroup comprised individuals without formal education, at 34.5% (n=76). Urban residents comprised 51.4% (n=113), and the most common income bracket was below 20,000 PKR per month, at 40.9% (n=90). Recurrent depression was reported by 75.0% (n=165) of the sample, with 69.1% (n=152) having no comorbidities. Over half of the participants had a disease duration of more than three years (53.2%, n=117), and the majority had more than one hospital visit per month (58.6%, n=129). The progressive phase was the most prevalent disease stage, affecting 62.7% (n=138) of the participants. The demographic and clinical characteristics are summarized in Table 1.

### Medication adherence of MDD patients

Table 2 summarizes medication adherence levels as measured by the UMGLS-4 and UDAI-10. According to the MGL, 39.1% (n=86) of participants displayed high medication adherence, 28.6% (n=63) demonstrated moderate adherence, and 32.3% (n=71) had poor adherence. In comparison, the DAI results indicated that 49.1% (n=108) of participants were adherent, 40.0% (n=88) showed moderate adherence, and 10.9% (n=24) were non-adherent. The trend is presented in Fig 1.

### Adherence scale translation and cultural adaptation

The expert committee conducted a content validity assessment of the UMGLS-4 in Urdu, focusing on semantic, idiomatic, experiential, and conceptual equivalence. They unanimously determined all items as "essential" for measuring medication adherence among Urdu-speaking MDD patients in Pakistan.

**Table 1. Demographic characteristics of MDD patients (N=220).**

| Variable | Category | Count | Percentage | p-Value* |
|---|---|---|---|---|
| **Gender** | Male | 111 | 50.5% | 0.762 |
| | Female | 109 | 49.5% | |
| **Age (Years)** | 18–29 | 42 | 19.1% | 0.026 |
| | 30–39 | 74 | 33.6% | |
| | 40–49 | 69 | 31.4% | |
| | ≥50 years | 35 | 15.9% | |
| **Marital Status** | Married | 124 | 56.4% | 0.206 |
| | Unmarried | 76 | 34.5% | |
| | Divorced | 18 | 8.2% | |
| | Others | 2 | 0.9% | |
| **Education** | Matriculation | 59 | 26.8% | 0.000 |
| | Intermediate | 27 | 12.3% | |
| | Graduation | 40 | 18.2% | |
| | Uneducated | 76 | 34.5% | |
| | Islamic Education | 18 | 8.2% | |
| **Occupational Status** | Employed | 53 | 24.1% | 0.001 |
| | Unemployed | 15 | 6.8% | |
| | Student | 48 | 21.8% | |
| | Housewife | 56 | 25.5% | |
| | Own Business | 42 | 19.1% | |
| | Retired | 6 | 2.7% | |
| **Residence** | Urban | 113 | 51.4% | 0.316 |
| | Rural | 107 | 48.6% | |
| **Monthly Income** | <20,000 | 34 | 15.5% | 0.047 |
| | 20,000–30,000 | 36 | 16.4% | |
| | 30,000–50,000 | 38 | 17.3% | |
| | 50,000–100,000 | 22 | 10.0% | |
| | >100,000 | 90 | 40.9% | |
| **Type of Depression** | Recurrent | 165 | 75.0% | 0.698 |
| | First Episode | 55 | 25.0% | |
| **Comorbidities** | Hypertension | 36 | 16.4% | 0.944 |
| | Diabetes | 13 | 5.9% | |
| | None | 152 | 69.1% | |
| | Others | 19 | 8.6% | |
| **Duration of Disease** | Less than 1 year | 70 | 31.8% | 0.440 |
| | 2–3 years | 33 | 15.0% | |
| | More than 3 years | 117 | 53.2% | |
| **Monthly Hospital Visits** | Once a month | 91 | 41.4% | 0.301 |
| | More than once | 129 | 58.6% | |
| **Disease Phase** | Initiation | 77 | 35.0% | 0.359 |
| | Progressive | 138 | 62.7% | |
| | Discontinued | 5 | 2.3% | |

*Simple linear regression identified significant ($p<0.05$, bolded) associations with MLG scale score

**Table 2. Medication adherence of MDD patients (N=220).**

| Medication Adherence | Level of Adherence | Count | Percentage |
|---|---|---|---|
| UMGLS-4 | High Medication Adherence | 86 | 39.1% |
| | Moderate Medication Adherence | 63 | 28.6% |
| | Poor Medication Adherence | 71 | 32.3% |
| UDAI-10 | Adherent | 108 | 49.1% |
| | Moderate Adherence | 88 | 40.0% |
| | Non-Adherence | 24 | 10.9% |

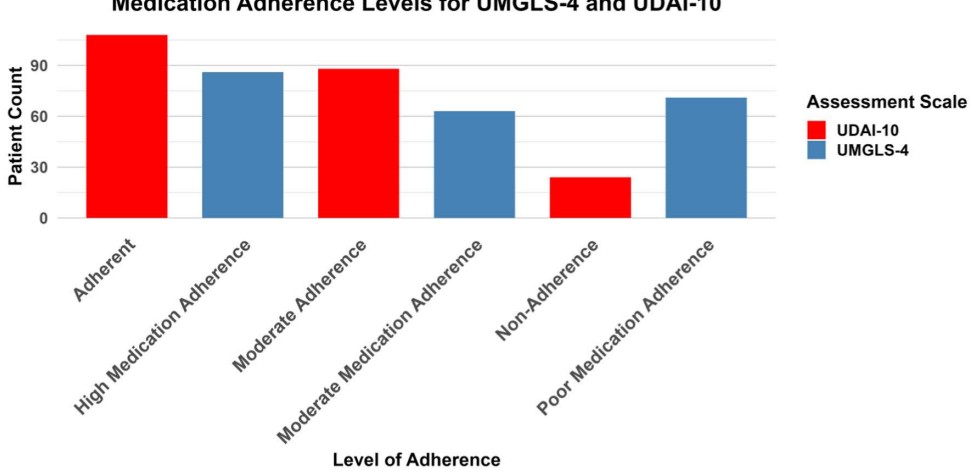

**Fig 1. Medication adherence of MDD Patients.**

The results of face validity showed that patients found the scale clear, understandable, and relevant to their experiences, confirming its face validity for assessing medication adherence in the Pakistani context.

### Reliability and test-retest reliability

The internal consistency of the medication adherence scale was assessed, revealing a Cronbach's α of 0.829, indicating a high degree of reliability. Each item's correlation with the total score was robust, with the deletion of any single item resulting in a Cronbach's α range between 0.772 and 0.796. These results are summarized in Table 3.

### Validity analyses

The convergent validity analysis between the UMGLS-4 and UDAI-10 scales demonstrated a moderate positive correlation (r = 0.622, $p < 0.01$), indicating a significant association between the scores on the two instruments.

The EFA through PCA revealed a single component with an eigen value >1, explaining 66.084% of the variance. The scree plot indicated a clear inflection point after the first component, further justifying the retention of a single-factor solution (Fig 2). The factor loadings for all items were above the commonly accepted threshold of 0.4, with values ranging from 0.791 to 0.834, demonstrating that each item had a strong loading on the principal component. These results collectively indicate a one-dimensional structure for the UMGLS-4, supporting its use as a reliable measure of medication adherence.

**Table 3. Reliability and internal consistency of UMGLS-4(N=220).**

| Item Description | Corrected Item-Total Correlation | Cronbach's *α* if Item Deleted |
|---|---|---|
| Do you often forget to take medicines? | 0.64 | 0.79 |
| Do you show carelessness in taking medicines? | 0.682 | 0.772 |
| Do you quit taking medicines to feel better? | 0.628 | 0.796 |
| Do you quit medicines when your disease gets worse? | 0.671 | 0.776 |
| **Cronbach's *α* (Overall)** | | 0.829 |

A significant strong ICC (x = 0.601, *p* <0.01) was detected, and a significant-excellent Pearson's correlation (r = 0.889, *p* < 0.01) was found between responses at time 1 and time 2 demonstrating good test-retest reliability.

The suitability of the data for CFA was confirmed by the Kaiser-Meyer-Olkin Measure of Sampling Adequacy, which yielded a value of 0.662, and Bartlett's Test of Sphericity, which was highly significant ($\chi^2$ = 370.872, df = 6, *p* < 0.001), indicating that the variables were sufficiently correlated for factor analysis. These results are summarized in Table 4.

## Medication adherence trends

Medication adherence continuous scores varied significantly across different demographics. Notably, males demonstrated a lower mean adherence score (M = 2.07, SD = ±1.62) compared to females (M = 2.55, SD = ±1.54), with this gender difference being statistically significant (*p* = 0.022). The examination of adherence by educational attainment revealed that individuals with only a matriculation level education had the lowest mean adherence score (M = 1.61, SD = ±1.56), which was significantly different from other educational levels (*p* = 0.001). Occupational status further distinguished adherence levels, where employed individuals had a mean score indicative of higher adherence (M = 1.72, SD = ±1.52) compared to their unemployed or student counterparts, presenting a statistically significant variation (*p* = 0.012). When considering the levels of high, moderate, and poor medication adherence categories, distinct patterns emerged. For gender, a higher proportion of males were found in the poor adherence category (62.0%, n = 44) than females (38.0%, n = 27), although the association was not statistically significant (*p* = 0.301). Educational levels correlated with adherence categories, where a significant proportion of individuals with matriculation were categorized under poor adherence (42.3%, n = 30), a statistically significant finding (*p* = 0.001). Additionally, employed individuals were more likely to be found in the high adherence category (27.0%, n = 17) as opposed to the poor adherence category (35.2%, n = 25), with this difference achieving statistical significance (*p* = 0.026). These results are presented in Table 5.

## Multivariate regression analyses

The model explains 17.22% of the variance in overall UMGLS score ($R^2$ = 0.1722), indicating moderate explanatory power. This level of explanatory power is typical in behavioural studies, as complex factors like socio-economic status, health literacy, and psychological factors influence it. The adjusted $R^2$ of 0.1408 shows the model is moderately fit after accounting for the predictors. The model is statistically significant (F = 5.487, *p* = 2.655e$^{-06}$), suggesting at least one predictor significantly influences medication adherence. The VIF values for all predictors (ranging from 1.18 to 1.44) indicate low multicollinearity, suggesting no need for corrective action in the model.

As shown in Table 6, the multivariate regression analysis revealed education as a significant predictor of MLG scale scores (B = 0.301, *p* < .000), indicating a strong association. Other variables, including gender and occupational status, showed positive relationships but did not reach statistical significance. Monthly income had minimal impact on UMGLS-4 scale scores

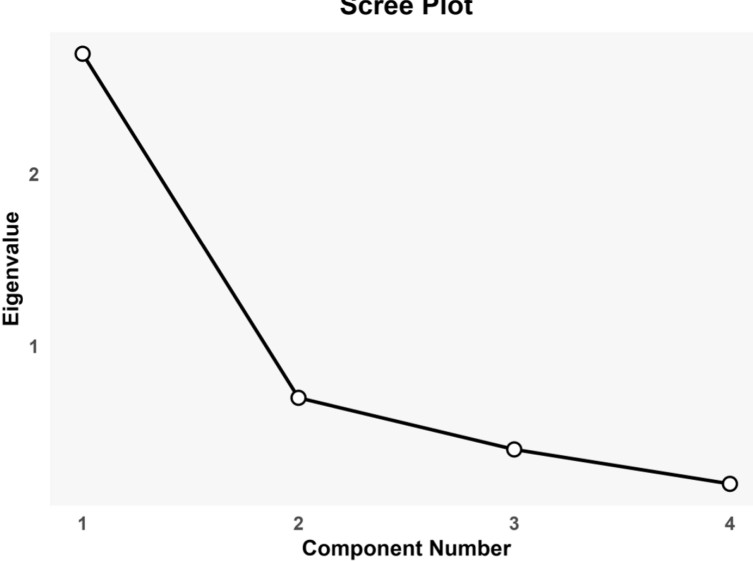

**Fig 2. Scree Plot showing Factor Loading for UMGLS-4.**

**Table 4. Exploratory Factor analyses of UMGLS-4.**

| Measure | Value |
|---|---|
| **KMO Measure of Sampling Adequacy** | 0.662 |
| **Bartlett's Test of Sphericity** | Approx. Chi-Square = 370.872, df = 6 <br> $p < 0.001$ |
| **Communalities** | **Extraction** |
| Item 1 | 0.646 |
| Item 2 | 0.695 |
| Item 3 | 0.626 |
| Item 4 | 0.677 |
| **Component Matrix** | **Factor Loading** |
| Item 1 | 0.804 |
| Item 2 | 0.834 |
| Item 3 | 0.791 |
| Item 4 | 0.823 |

CFA supported a satisfactory fit for the proposed scale, with all indices falling within acceptable ranges: GFI = 0.950, AGFI = 0.920, NFI = 0.930, RMSEA = 0.050, and SRMSR = 0.055. These results indicate a good model fit to the observed data.

## Discussion

Medication adherence is crucial for managing conditions such as MDD in culturally diverse regions like Pakistan. This study aimed to translate, culturally adapt, and psychometrically evaluate the UMGLS-4 among MDD patients in Pakistan. Our findings indicate that the UMGLS-4 is a reliable and valid instrument for assessing medication adherence in Urdu-speaking patients with MDD. The Urdu translated and adopted version of MGLS-4 is available as supporting information (S1). The Cronbach's *a* value of UMGLS-4 was 0.829, showcasing a strong internal consistency [51]. This is

**Table 5. Medication adherence trend among MDD patients(N=220).**

| Variable | Category | Mean (± S.D) | p-Value* | High Medication Adherence (n, %) | Moderate Medication Adherence (n, %) | Poor Medication Adherence (n, %) | p-Value** |
|---|---|---|---|---|---|---|---|
| **Gender** | Male | 2.07 (±1.62) | **0.022** | 35 (40.7%) | 32 (50.8%) | 44 (62.0%) | 0.301 |
| | Female | 2.55 (±1.54) | | 51 (59.3%) | 31 (49.2%) | 27 (38.0%) | |
| **Age (Years)** | 18–29 | 2.38 (±1.72) | 0.844 | 20 (23.3%) | 8 (12.7%) | 14 (19.7%) | 0.624 |
| | 30–39 | 2.16 (±1.66) | | 28 (32.6%) | 20 (31.7%) | 26 (36.6%) | |
| | 40–49 | 2.39 (±1.47) | | 24 (27.9%) | 25 (39.7%) | 20 (28.2%) | |
| | ≥50 years | 2.37 (±1.59) | | 14 (16.3%) | 10 (15.9%) | 11 (15.5%) | |
| **Marital Status** | Married | 2.23 (±1.57) | 0.342 | 44 (51.2%) | 38 (60.3%) | 42 (59.2%) | 0.307 |
| | Unmarried | 2.33 (±1.61) | | 30 (34.9%) | 23 (36.5%) | 23 (32.4%) | |
| | Divorced | 2.56 (±1.75) | | 10 (11.6%) | 2 (3.2%) | 6 (8.5%) | |
| | Others | 2.03 (±1.23) | | 2 (2.3%) | 0 (0.0%) | 0 (0.0%) | |
| **Education** | Matriculation | 1.61 (±1.56) | **0.001** | 13 (15.1%) | 16 (25.4%) | 30 (42.3%) | **0.001** |
| | Intermediate | 1.93 (±1.61) | | 8 (9.3%) | 6 (9.5%) | 13 (18.3%) | |
| | Graduation | 2.40 (±1.49) | | 15 (17.4%) | 15 (23.8%) | 10 (14.1%) | |
| | Uneducated | 2.84 (±1.46) | | 41 (47.7%) | 21 (33.3%) | 14 (19.7%) | |
| | Islamic Education | 2.72 (±1.63) | | 9 (10.5%) | 5 (7.9%) | 4 (5.6%) | |
| **Occupational Status** | Employed | 1.72 (±1.52) | 0.012 | 11 (12.8%) | 17 (27.0%) | 25 (35.2%) | **0.026** |
| | Unemployed | 2.40 (±1.45) | | 5 (5.8%) | 6 (9.5%) | 4 (5.6%) | |
| | Student | 2.21 (±1.71) | | 20 (23.3%) | 11 (17.5%) | 17 (23.9%) | |
| | Housewife | 2.59 (±1.49) | | 25 (29.1%) | 19 (30.2%) | 12 (16.9%) | |
| | Own Business | 2.83 (±1.57) | | 24 (27.9%) | 8 (12.7%) | 10 (14.1%) | |
| | Retired | 1.83 (±1.47) | | 1 (1.2%) | 2 (3.2%) | 3 (4.2%) | |
| **Residence** | Urban | 2.20 (±1.59) | 0.299 | 40 (46.5%) | 36 (57.1%) | 37 (52.1%) | 0.434 |
| | Rural | 2.42 (±1.60) | | 46 (53.5%) | 27 (42.9%) | 34 (47.9%) | |
| **Monthly Income** | <20,000 | 1.91 (±1.56) | 0.308 | 9 (10.5%) | 12 (19.0%) | 13 (18.3%) | 0.52 |
| | 20,000–30,000 | 2.64 (±1.59) | | 11 (12.8%) | 10 (15.9%) | 15 (21.1%) | |
| | 30,000–50,000 | 2.26 (±1.68) | | 15 (17.4%) | 9 (14.3%) | 14 (19.7%) | |
| | 50,000–100,000 | 2.48 (±1.58) | | 10 (11.6%) | 6 (9.5%) | 6 (8.5%) | |
| | >100,000 | 2.37 (±1.31) | | 41 (47.7%) | 26 (41.3%) | 23 (32.4%) | |
| **Type of Depression** | Recurrent | 2.33 (±1.59) | 0.712 | 65 (75.6%) | 48 (76.2%) | 52 (73.2%) | 0.914 |
| | First Episode | 2.24 (±1.64) | | 21 (24.4%) | 15 (23.8%) | 19 (26.8%) | |
| **Comorbidities** | Hypertension | 2.14 (±1.58) | 0.412 | 12 (14.0%) | 10 (15.9%) | 14 (19.7%) | **0.001** |
| | Diabetes Mellitus | 2.62 (±1.26) | | 4 (4.7%) | 7 (11.1%) | 2 (2.8%) | |
| | None | 2.37 (±1.69) | | 70 (81.4%) | 32 (50.8%) | 50 (70.4%) | |
| | Others | 1.59 (±0.97) | | 0 (0.0%) | 14 (22.2%) | 5 (7.0%) | |
| **Duration of Disease** | <1 year | 2.33 (±1.55) | 0.066 | 26 (30.2%) | 24 (38.1%) | 20 (28.2%) | 0.129 |
| | 2–3 years | 1.73 (±1.58) | | 8 (9.3%) | 9 (14.3%) | 16 (22.5%) | |
| | >3 years | 2.46 (±1.60) | | 52 (60.5%) | 30 (47.6%) | 35 (49.3%) | |
| **Monthly Hospital Visits** | Once a month | 2.18 (±1.62) | 0.289 | 32 (37.2%) | 29 (46.0%) | 30 (42.3%) | 0.548 |
| | More than once | 2.40 (±1.59) | | 54 (62.8%) | 34 (54.0%) | 41 (57.7%) | |
| **Disease Phase** | Initiation | 2.14 (±1.58) | 0.426 | 26 (30.2%) | 24 (38.1%) | 27 (38.0%) | 0.383 |
| | Progressive | 2.41 (±1.61) | | 59 (68.6%) | 36 (57.1%) | 43 (60.6%) | |
| | Discontinued | 2.00 (±1.41) | | 1 (1.2%) | 3 (4.8%) | 1 (1.4%) | |

*Kruskal Wallis/ Mann Whitney Tests ($p < 0.005$ was considered Significant)

**Chi-square/ Fisher Exact test ($p < 0.05$ was considered Significant)

**Table 6. Results from multivariate regression analyses (N=220).**

| Variables | Unstandardized Coefficients | | Standardized Coefficients | t | Sig. | VIF |
|---|---|---|---|---|---|---|
| | B | Std. Error | Beta | | | |
| (Constant) | 0.432 | 0.407 | | 1.061 | 0.290 | |
| Gender | 0.312 | 0.219 | 0.098 | 1.426 | 0.155 | 1.20 |
| Education | 0.301 | 0.079 | 0.256 | 3.815 | **0.000** | 1.26 |
| Occupational status | 0.138 | 0.073 | 0.130 | 1.899 | 0.059 | 1.44 |
| Monthly Income | 0.034 | 0.074 | 0.032 | 0.462 | 0.644 | 1.18 |

higher than the values reported by Tzeng et al. (2008) for Taiwanese cancer pain patients (0.73), Kristina et al. (2019) for Indonesian DM patients (0.651), Wang et al. (2012) for Singaporean Diabetes Type 2 patients (0.62), and Morisky et al. (1986) for English-speaking hypertension (HTN) population (0.61), indicating a more reliable adherence measurement in this context [19,23–25]. However, it is slightly lower than Elhenawy et al. (2022) who reported a 0.857 Cronbach's $\alpha$ in an Arabic-speaking DMT2 context [21], and Lu et al. (2016) who reported 0.857 in Chinese MDD patients [54], and higher than Awwad et al. (2022) with 0.593 in Arabic-speaking patients with chronic conditions [20]. The diversity in translation languages and disease conditions across these studies underscores the broad spectrum of adherence research in varied cultural and medical contexts.

The current study had a sample size of 220, which is moderate when compared across the board. It is larger than Tzeng et al. (2008) with 135 patients and Awwad et al. (2022) with 201 patients but smaller than Kristina et al. (2019) with 250 patients, Wang et al. (2012) with 294 patients, and significantly smaller than Elhenawy et al. (2022) who utilized a sample of 400 patients [20,21,23–25].

The convergent validity in our study was reported at r = +0.622, which is higher than the r=0.50 and 0.45 reported by Tzeng et al. (2008) and comparable to the r=0.58 by Kristina et al. (2019) [23,24]. However, it is lower than the r=0.69 reported by Elhenawy et al. (2022) and significantly lower than the r=0.742 by Awwad et al. (2022) [20,21]. This reflects variations in the measurement tools' efficacy in capturing adherence across different health conditions and populations.

Our study identified significant gender differences in adherence scores, with males displaying lower adherence than females, a statistically significant disparity. This finding is consistent with the broader literature, including reports of suboptimal adherence among Chinese MDD patients, where 62.2% were not fully adherent, often due to forgetfulness or perceived health improvement [54]. Similarly, Indian MDD studies highlighted the influence of education, drug number, and family income on adherence, reflecting our observation that educational attainment significantly affects adherence levels [55,56]. While this result was not directly hypothesized, it suggests that higher education may contribute to better understanding and management of medical regimens, potentially due to improved health literacy or better access to resources [57]. Additionally, employment status was a determinant of adherence in our cohort, with employed individuals showing higher adherence (M = 1.72, SD = +1.52) than their unemployed or student counterparts, marking a statistically significant difference (p = 0.012). The influence of employment on medication adherence might be due to factors such as better healthcare access, structured routines, and financial stability [38]. Despite the lack of statistical significance in gender differences within adherence categories, the clear influence of educational and employment status on adherence patterns aligns with findings from previous studies that reported good adherence in 72.4% of their study participants [58].

The multivariate regression of present study highlights that education significantly influences medication adherence, underscoring its pivotal role in medication adherence among the variables studied. This finding is similar to previous studies [59]. Medication adherence can be improved by providing educational sessions to MDD patients, however, one-time educational intervention is not helpful [60]. These insights suggest that the UMGLS-4 could guide interventions to improve adherence, particularly through educational sessions and targeted support for at-risk patients for MNA.

The results of this study highlight that clinician can use the UMGLS-4 to identify patients with low adherence and tailor interventions accordingly. For patients exhibiting unintentional MNA, strategies such as reminder systems or adjustments to the medication regimen could be implemented [61]. In cases of intentional nonadherence, a more personalized approach, such as counseling to address concerns about treatment, would likely be more effective [62]. Additionally, the 4-item nature of the UMGLS-4 makes it a quick and efficient tool for screening, minimizing the time burden on clinicians while still providing valuable insights into medication adherence [63]. By integrating the UMGLS-4 into clinical practice, healthcare providers can better monitor adherence and improve patient outcomes [19]. By integrating the UMGLS-4 into clinical practice, healthcare providers can monitor adherence and improve patient outcomes.

This study is not without limitations. Although, convenience sampling was used in this study due to its efficiency and cost-effectiveness in recruiting participants from easily accessible OPD. This sampling method and the study's setting in an urban psychiatric OPD may limit the generalizability of our findings to the broader Pakistani population, particularly those in rural areas or with different healthcare access levels. Additionally, individuals from lower socioeconomic backgrounds may experience challenges like financial instability or stigma that can influence their adherence behaviors in unique ways, which may not be reflected in a sample from a more urban, middle-to-high-income population [64]. Moreover, the cross-sectional design precludes the assessment of the scale's sensitivity to changes over time, an aspect that future longitudinal studies could explore. Furthermore, the KMO index of 0.662 indicates moderate sampling adequacy, which may limit the robustness and interpretability of the factor analysis results.

Future research should aim to validate the UMGLS-4 in other diseased conditions in the Pakistani population. Longitudinal studies are also needed to evaluate the scale's responsiveness to changes in medication adherence over time, which could further establish its utility in monitoring and guiding interventions. Investigating the relationship between medication adherence, as measured by the UMGLS-4, and clinical outcomes in MDD patients could provide deeper insights into its predictive validity and the potential impact of adherence-enhancing interventions.

## Conclusions

This study's validation and reliability of the UMGLS-4 for MDD patients in Pakistan marks a significant step toward addressing medication adherence screening for clinical and research purposes. The scale offers a quick and effective method for detecting nonadherence, ideal for use in busy clinical environments. The research highlights key demographic factors—gender, education, and occupation—that significantly influence adherence levels, aligning with global trends in MDD management. These insights not only underscore the importance of targeted interventions to improve adherence but also pave the way for further studies on UMGLS-4's broader application. Integrating UMGLS-4 into routine clinical practice allows healthcare providers to identify at-risk patients and improve MDD outcomes through enhanced adherence strategies,

## Supporting information

**S1 Fig. Urdu version of Morisky, Greene, and Levine Medication Adherence Scale (UMGLS-4).**
(PDF)

**S1 Data. Urdu version of MGLS-4 (UMGLS-4).**
(XLSX)

## Acknowledgments

We are grateful to Dr. Donald E. Morisky for allowing the use of the MGLS-4 and Dr. Fahad Saleem, University of Balochistan, for his kind permission to utilize the UDAI-10 for our study.

## Author contributions

**Conceptualization:** Fazli Khuda, Sohail Riaz, Sajid Ali, Asif Jan.

**Data curation:** Nitasha Gohar, Aqeel Nasim.

**Formal analysis:** Ayesha Rashid, Asif Jan.

**Investigation:** Sohail Riaz.

**Methodology:** Fazli Khuda, Sohail Riaz, Abuzar Khan.

**Supervision:** Fazli Khuda, Abuzar Khan.

**Writing – original draft:** Sohail Riaz, Nadia Shamshad Malik, Nitasha Gohar, Abdur Rahman.

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
