## [Decision Letter · Decision Letter 0]

2 Jan 2025

PONE-D-24-26034Cross-Cultural Adaptation and Psychometric Evaluation of the Urdu Version of the Morisky, Greene, and Levine Medication Adherence Scale (MGLS-4) for Major Depressive Disorder PatientsPLOS ONE

Dear Dr. Khuda,

Thank you for submitting your manuscript to PLOS ONE. After careful consideration, we feel that it has merit but does not fully meet PLOS ONE’s publication criteria as it currently stands. Therefore, we invite you to submit a revised version of the manuscript that addresses the points raised during the review process. Please submit your revised manuscript by Feb 16 2025 11:59PM. If you will need more time than this to complete your revisions, please reply to this message or contact the journal office at plosone@plos.org . Please include the following items when submitting your revised manuscript:

We look forward to receiving your revised manuscript.

Kind regards,

Marco Innamorati

Academic Editor

PLOS ONE

Journal Requirements:

**Additional Editor Comments:**

Dear Authors,

We have now reports from reviewers. They suggested revisions for your manuscript. Please, revise your manuscript according to their suggestions.

Reviewers' comments:

Reviewer's Responses to Questions

**Comments to the Author**

1. Is the manuscript technically sound, and do the data support the conclusions?

Reviewer #1: Partly

Reviewer #2: Yes

2. Has the statistical analysis been performed appropriately and rigorously? 

Reviewer #1: No

Reviewer #2: Yes

3. Have the authors made all data underlying the findings in their manuscript fully available?

Reviewer #1: Yes

Reviewer #2: Yes

4. Is the manuscript presented in an intelligible fashion and written in standard English?

Reviewer #1: Yes

Reviewer #2: No

5. Review Comments to the Author

Reviewer #1: Introduction

The introduction is well-structured, giving a clear overview of the problem. However, it would be beneficial if there were closer links between the need for the cultural and linguistic validation of UMGLS-4 and the clinical challenges observed in Pakistan. Emphasizing, for instance, the potential practical implications of the validated scale could make the introduction more appealing.

Objectives

Objectives are clear and pertinent. However, elaborating on the clinical significance of validating the UMGLS-4 would amplify its relevance. For instance, discussing how this tool would be utilized in a clinical setup for advancing medication adherence strategies for MDD patients.

Methodology

It is well-articulated, and the various steps taken regarding translation and cultural adaptation have been adequately detailed. Nevertheless, several features have the scope for modification or fine-tuning:

1. Sampling: Convenience sampling is used, which is fine, but this should be discussed in a more elaborative manner, for instance, how would this limit the generalization of the findings in rural or socioeconomically diverse populations.

2. Translation Process: The elaborate translation and back-translation process is really appreciable.

3. Statistical Analyses: The application of non-parametric tests, such as Kruskal-Wallis and Mann-Whitney U, is appropriate but requires justification. Were assumptions of normality and homoscedasticity tested, for example, the Shapiro-Wilk test? This would enhance the appropriateness of the statistical method.

4. Factor Analysis: Although EFA and CFA were performed, the Kaiser-Meyer-Olkin (KMO) index of 0.662 indicates only moderate sampling adequacy. Limitations of the study need to be declared explicitly.

Results

The results are comprehensive and well-organized. However, some areas could be presented more effectively:

1. Medication Adherence Trend: The demographic variations, though explained, would still be easier to understand when summarized using visual aids (such as bar graphs or pie charts) that highlight their key differences. The observed major association with education is highly intriguing but the result does not relate it to any hypothesis. What could explain why education will make a difference here? Also, influence from employment status on adherence must also be explored further.

2.Regression Analysis: The multivariate regression analysis is useful, but key metrics such as the coefficient of determination ( R^2 ) and the overall model significance test (F-test) are missing.Including these would provide a clearer picture of the model’s explanatory power. While education emerges as a significant predictor, the lack of significance for other variables (e.g., gender, occupation) raises questions. Could collinearity or insufficient sample size have affected these results? Including the VIF would complete the concern of multicollinearity. Discussion

The discussion is well-expressed and puts the findings into the perspective of the existing literature. However, it has a heavy leaning towards comparing findings with those of previous studies. There is a need for deeper reflection on what the findings mean in terms of practice. For example, how might clinicians or policy makers take this tool and use it to enhance adherence in similar settings?

Conclusions

The conclusions are well-written, but could be stronger in driving home the usefulness of the UMGLS-4 by explicitly discussing its potential role in guiding interventions to improve

Reviewer #2: This manuscript presents a study conducted by Khuda and colleagues, aimed to translate into Urdu and validate the Morisky, Greene, and Levine Medication Adherence Scale (MGLS-4) in patients with Major Depressive Disorder (MDD). Some issues need to be addressed.

Throughout the manuscript, there are several typographical errors: periods following commas (e.g., line 57; p.8), spaces before periods (e.g., line 110; p.12), extra spaces (e.g., lines 159, 179, 199; pp. 14, 15, 16 respectively), commas instead of periods (e.g., line 210; p.16), and missing spaces (e.g., line 214; p.16). Furthermore, the writing style should be standardized. For example, p-values are italicized in some parts of the manuscript but not in others. It would be better to italicize all p-values, and it is not necessary to specify the term "value" in brackets, you could simply write (p=/...).

When presenting suggestions without references to support the statements, it wold be preferable to use conditional verbs (e.g., lines 97-98; p.10: “[…] it could be crucial”).

Abstract & introduction: these sections are well structured and fluent.

- In the abstract, the term “behaviour” appears (line 41; p.8), while elsewhere in the manuscript the term “behavior(s)” is used. I recommend standardizing the text to a consistent linguistic form, and reviewing whether other terms also need standardization.

- In the introduction, I suggest providing a reference and page number for the citation reported in lines 84-85 (p. 10). Additionally, in line 86 (p. 11) the second occurrence of “Medication nonadherence” can be omitted, leaving only the acronym, as the term has already been introduced in full earlier. It would also be helpful to clarify or rewrite line 89 (p. 11), when there is reported “This difference”, it does not make clear what difference is being referenced.

Methodology: this section is well articulated, however some additional information is needed.

- In the “Study population and sampling” paragraph, I suggest explicitly stating that the inclusion and exclusion criteria are been reported (lines from 130 to 133; pp. 12-13). Furthermore, it would be preferable to provide more details regarding the sampling methods, such as who carried out the recruitment, how it was conducted (e.g., in-person or by telephone), when patients were recruited (e.g., during pre-scheduled visits or during their hospitalizations).

- In the “Translation and Cultural Adaptation of MGLS-4 into Urdu” paragraph, I recommend writing out the full name of the tool and including the acronym in parentheses (line 139; p. 13).

- In the “Data collection and questionnaires” paragraph, specify how the data were collected (e.g., via online systems or paper forms - in the latter case, who supervised the process). Additionally, I suggest reporting the Cronbach’s alpha of your study for the DAI-10 to ensure reliability is clearly communicated.

- In the “Statistical analyses” paragraph, I suggest simplifying lines 179-180 (p.14) writing “demographic and clinical characteristics, and …”.

Results: this section thoroughly reports all data analyses, in line with the previous paragraph

In line 229 (p. 18) I suggest removing the period after “2” (Table 2.) and deleting the final sentence, as the table is already presented above.

In line 304 (p.26), I suggest including the zero before the decimal in B-value (i.e., B=0.301).

Discussion & conclusion: these sections are well written. I suggest beginning the Discussion with one or two brief sentences summarizing the background of the study.

6. PLOS authors have the option to publish the peer review history of their article (what does this mean? ). If published, this will include your full peer review and any attached files.

**Do you want your identity to be public for this peer review?** For information about this choice, including consent withdrawal, please see our Privacy Policy .

Reviewer #1: No

Reviewer #2: No

---

## [Author Response · Author response to Decision Letter 1]

16 Jan 2025

Please find attached a rebuttal letter.

---

## [Decision Letter · Decision Letter 1]

16 Feb 2025

Cross-Cultural Adaptation and Psychometric Evaluation of the Urdu Version of the Morisky, Greene, and Levine Medication Adherence Scale (MGLS-4) for Major Depressive Disorder Patients

PONE-D-24-26034R1

Dear Dr. Khuda,

We’re pleased to inform you that your manuscript has been judged scientifically suitable for publication and will be formally accepted for publication once it meets all outstanding technical requirements.

Kind regards,

Marco Innamorati

Academic Editor

PLOS ONE

Additional Editor Comments (optional):

Reviewers' comments:

Reviewer's Responses to Questions

**Comments to the Author**

1. If the authors have adequately addressed your comments raised in a previous round of review and you feel that this manuscript is now acceptable for publication, you may indicate that here to bypass the “Comments to the Author” section, enter your conflict of interest statement in the “Confidential to Editor” section, and submit your "Accept" recommendation.

Reviewer #1: All comments have been addressed

Reviewer #2: All comments have been addressed

2. Is the manuscript technically sound, and do the data support the conclusions?

Reviewer #1: (No Response)

Reviewer #2: Yes

3. Has the statistical analysis been performed appropriately and rigorously? 

Reviewer #1: (No Response)

Reviewer #2: Yes

4. Have the authors made all data underlying the findings in their manuscript fully available?

Reviewer #1: (No Response)

Reviewer #2: Yes

5. Is the manuscript presented in an intelligible fashion and written in standard English?

Reviewer #1: (No Response)

Reviewer #2: Yes

6. Review Comments to the Author

Reviewer #1: (No Response)

Reviewer #2: (No Response)

7. PLOS authors have the option to publish the peer review history of their article (what does this mean? ). If published, this will include your full peer review and any attached files.

**Do you want your identity to be public for this peer review?** For information about this choice, including consent withdrawal, please see our Privacy Policy .

Reviewer #1: No

Reviewer #2: No

---

## [Editor Report · Acceptance letter]

PONE-D-24-26034R1

PLOS ONE

Dear Dr. Khuda,

I'm pleased to inform you that your manuscript has been deemed suitable for publication in PLOS ONE. Congratulations! Your manuscript is now being handed over to our production team.

Kind regards,

on behalf of

Dr. Marco Innamorati

Academic Editor

PLOS ONE